# Would Kazakh Citizens Support a Milk Co-Operative System?

**Samal Kaliyeva** [1,*], **Francisco Jose Areal** [2] and **Yiorgos Gadanakis** [1]

1 School of Agriculture Policy and Development, University of Reading, Reading RG6 6AR, UK; g.gadanakis@reading.ac.uk

2 Centre for Rural Economy, School of Natural and Environmental Sciences, Newcastle University, Newcastle upon Tyne NE1 7RU, UK; francisco.areal-borrego@newcastle.ac.uk

* Correspondence: s.kaliyeva@pgr.reading.ac.uk

**Abstract:** We estimate the monetary value of a policy aimed at increasing rural co-operative production in Kazakhstan to increase milk production. We analyse the drivers associated with public support for such policy using the contingent valuation method. The role of individuals' psychological aspects, based on the reasoned action approach, along with individuals' views on the country's past regime (i.e., to the former Soviet Union), their awareness about the governmental policy, their sociodemographic characteristics, and household location on their willingness to pay (WTP) for the policy is analysed using an interval regression model. Additionally, we examine changes in individuals' WTP before and during the COVID-19 pandemic. The estimated total economic value of the policy is KZT 1335 bn for the length of the program at KZT 267 bn per year, which is approximately half the total program budget, which includes other interventions beyond the creation of production co-operatives. The total economic value of the policy would equal the cost of the whole program after 10 years, indicating public support for this policy amongst Kazakh citizens. Psychological factors, i.e., attitude, perceived social pressure, and perceived behavioural control, and the respondents' awareness of the policy and views on the Soviet Union regime are associated with their WTP. Sociodemographic factors, namely, age, income, and education, are also statistically significant. Finally, the effect of the shocks of COVID-19 is negatively associated with the respondents' WTP.

**Keywords:** co-operative creation policy; contingent valuation; reasoned action approach; Kazakhstan; COVID-19

## 1. Introduction

Prior to Kazakhstan joining the World Trade Organisation in 2015, Kazakhstan joined Belarus and Russia in 2014 to create the Eurasian Economic Union (EAEU), a free trade zone. Later Armenia, Belarus, and Kyrgyzstan also joined the EAEU. The opening of Kazakhstan's economy to international markets challenged its agricultural competitiveness, which was detrimental to the rural economy, highly dependent on agricultural production [1]. Hence, improving agriculture productivity is key for the development of the rural economy of Kazakhstan. Consequently, the government decided to stimulate the production of agricultural products by allocating a significant part of its governmental budget, 2374.2 billion tenges (KZT) for 5 years, for the development of the country's agricultural sector, part of which also considers the creation of agricultural co-operatives. This is a relatively large budget, accounting for 9% of the revenue of the state, republican, and local budgets in 2017. To compare, 1868.4 billion tenges (KZT) was budgeted under the state program for the development of education and science for the period 2016–2019; 1385.6 billion was budgeted for the development of tourism for the period 2019–2025, and 1762.5 billion tenges was budgeted for regional development for the period 2015–2020 [2].

Amongst agricultural products produced in Kazakhstan, dairy is one of the key agricultural sectors, representing 16% of the total agricultural production of the country [3]. Milk production has increased by 16% in the last 5 years reaching a total of 5,820,000 t

of cow's milk produced in 2019 compared to 5,020,000 t in 2014. However, the domestic supply of dairy products is insufficient to meet the internal demand. Specifically, dairy product exports amounted to USD 53,517,500 (1 US dollar (USD) is equal to 426.84 tenges (KZT) as of 27 June 2021), whereas the imports were USD 252,450,400 in 2019, indicating a 198,932,900 trade deficit. Hence, a transformation of the structure of the dairy sector seems key to reduce this gap.

Currently, the structure of Kazakh's dairy is dominated by small-scale producers, such as rural households and individual/peasant farms, representing 93% of total production (of which rural households are 78% and individual/peasants are 22%), whereas only 7% of the milk was produced by agricultural enterprises. Thus, due to the prevailing of small-scale production, dairy factories face a deficit of milk for processing, and consequently, the country experiences a low supply of processed dairy products [4,5]. In 2019, a total of 262,000 t of milk went to the processing factories in Kazakhstan, only 4.5% of the 5,820,000 t produced that year. Considering therefore the status of the agricultural sector, the government's intervention plan aimed at reducing the number of agricultural activities conducted by small farm/household with the objective of expanding agricultural production (including dairy) in enterprises through the creation of co-operatives in rural areas. It is worth noting that although there are other supply chain pathways to reach dairy factories (e.g., peasant and small farms, merchants), more than 70% of milk is produced by rural households, consequently making them the main body in the dairy supply chain.

The legislative basis of co-operatives is set out in the law "On Agricultural Cooperation", adopted in 2015. The policy on creating co-operatives was introduced in 2017. However, the initial government plan was revised in July 2018, and is no longer aiming to create more co-operatives under the Programme (the reason of which remains unclear). Despite this fact, the idea of creating co-operatives is still relevant and it has been included in the Strategic Plan of noncommercial organization "Atameken" for 2018–2023; thus, in 2019, the number of rural households involved in co-operative production was 27.2 thousand whereas the production of cow's milk by co-operatives was 65.4 thousand tonnes (the country's total production was 5820.1 thousand tonnes of milk in the same year).

According to the law, an agricultural co-operative is created when there are at least three members. All members of the co-operative are obliged to pay an entrance fee, in accordance with the charter of the co-operative. If necessary, members of the co-operative can make additional contributions (on a voluntary basis). In addition, the founders and members of the co-operative can also make a material (share) contribution. The basic principles of the creation and functioning of co-operatives are expected to comply with the international principles specified in the International Co-operative Alliance (ICA). According to the ICA, there are seven main international co-operative principles: (1) voluntary and open membership; (2) democratic member control; (3) member economic participation; (4) autonomy and independence; (5) provision of education, training, and information; (6) co-operation among co-operatives; and (7) concern for the community.

Unlike the Soviet Union where production output and all assets (productive and social, except land) were owned jointly by the collective (i.e., kolkhozes) and by the state (i.e., sovkhozes/state farms) [6], under the current policy, the individuals do not own the means of production and share the means of production to produce an output. Nevertheless, access to technologies, equipment, feeding, and subsidies are expected to be facilitated through co-operatives.

Although co-operatives can potentially be organised in many forms, e.g., service co-operatives, the main focus of the policy and therefore of this study is focused on production co-operatives. Rural households are expected to be engaged in the supply chain to facilitate constant milk supply to dairy factories via co-operatives. Members of production co-operatives, i.e., rural households and individual/peasant farms, are expected to supply the co-operatives with fresh milk that goes directly to the dairy processing industry. As there are no intermediates, rural households (and individual/peasant farms) will be paid from

the dairy processing units directly. In turn, co-operatives receive KZT 10 per litre of milk in the form of subsidies from the government [7].

Co-operatives can contribute to uplifting livelihoods by reducing poverty and food insecurity in rural areas through the improved use of technology, share of knowledge between members, and distribute income from a market-oriented output [8–11].

We estimate the consumer's willingness to pay (WTP) for a Kazakh's government intervention to create production co-operatives in rural areas to obtain the total economic value of the policy. We also analyse the heterogeneity in WTP and investigate whether the COVID-19 pandemic affected consumer WTP for the government's policy. Estimating the total economic value of agricultural policies, or any other policy for that matter, is paramount for policy decision-making under constrained budgets. As Price [12] points out, an "unbiased and focused evaluation of unpriced benefits is an important pre-condition for needed policy interventions". The estimation of monetary value of agricultural policies, such as conservation of agricultural genetic resources [13], safe vegetables [14], and agri-environment schemes [15] has been previously studied. Although the attitudes of Kazakh rural households towards joining and creating co-operatives was previously studied [16], to the best of our knowledge, no study has estimated the total economic value of a policy aimed at increasing milk production through co-operative creation. More specifically, we contribute to the literature in three ways: (1) by estimating the total economic value a of the transformation of the milk production system from small-scale production to industrial production through a policy aiming at creating co-operatives; (2) to our knowledge, this is the first paper that has used and expanded the reasoned action approach to gaining an understanding of how the total economic value for the policy is moderated by a number of elements. These include individual psychological aspects based on the reasoned action approach (RAA), views on the past regime (i.e., to the former Soviet Union), awareness concerning the governmental policy, sociodemographic characteristics, and geographical location; and (3) by analysing whether a pandemic shock such as COVID-19 may be associated with changes in individuals' WTP for the policy.

## 2. Materials and Methods

We used the contingent valuation (CV) method to elicit the total economic value of the policy through the respondents' WTP for a premium price on a litre of milk in order to support the government policy. The program allows farmers to receive support from government and other co-operatives, such as a subsidy in the amount of KZT 10 per litre of milk and discounted animal feed products. This information was provided to respondents along with the policy objective of supporting dairy producing households to expand dairy production in Kazakhstan. We used the RAA to analyse how psychological factors may be associated to respondents' WTP. We extend the RAA to integrate the respondents' (a) views on the past regime (i.e., to the former Soviet Union), (b) their sociodemographic characteristics and the location, (c) awareness about the governmental policy, and (d) COVID-19 into our framework to investigate the role of these elements on respondents' WTP.

### 2.1. Contingent Valuation Method

The total value associated with the implementation of governmental policies includes not only the provision of market goods, but the provision of nonmarket goods and services, too (i.e., those that cannot be traded in the marketplace, and consequently do not have a market price). The policy might provide substantial benefits for the society, such as increasing milk production whilst supporting rural development and allowing farmers to increase their livelihoods as a result of receiving higher returns for their products. Co-operative production promotes sustainable agriculture, enhancing not only the environment but also the social sustainability of local communities [17]. The stated preferences method is employed as a double-bounded dichotomous choice contingent valuation (CV) to elicit the total value of the policy. Although the majority of the stated preference research focuses on

the demand for environmental benefits, the use of this technique has spread to evaluating other type of goods, including farmers' WTP for crop insurance [18], animal welfare [19], agricultural genetic resources [13], and the provision of production services [20].

Preferences of the respondents are explained by the random utility theory (RUT) since it is the theoretical basis for the CV method [21,22]. Thus, the utility of a good is expressed as follows:

$$U_{iq} = V_{iq} + \varepsilon_{iq} \tag{1}$$

where $U$ is the utility of good $i$ for individual $q$, $V_{iq}$ is the expected value of $U$, and $\varepsilon$ is the error term.

Two main approaches are used to elicit the value of a good using CV: (a) single-bounded (take-it-or-leave-it) and (b) double-bounded (take-it-or-leave-it with follow-up) dichotomous choice techniques. However, the single-bounded approach has been criticized due to the limitation in revealing the true WTP [23,24]. The double-bounded dichotomous choice approach was used to deal with the limitations of a single-bound approach. The singularity of this approach is that participants are simply asked if they would pay a certain amount of money for the good and if the answer is "Yes" ("No"), the monetary amount can be raised (or decreased) with follow-up questions according to Yes/No answers [23,25–27]. Consequently, by follow-up questions, four possible outcomes can be derived [28]:

1.　Respondent answers YES for both the main bid $P^I$ and the higher bid $P^H$ (YES–YES), in this case, WTP $\geq P^H$
2.　Respondent answers YES for the main bid $P^I$ and NO for the higher bid $P^H$ (YES–NO), in this case, $P^I \leq$ WTP $> P^H$
3.　Respondent answers NO for the main bid $P^I$ and YES for the lower bid $P^L$ (NO–YES), in this case, $P^L \leq$ WTP $< P^I$
4.　Respondent answers NO for both the main bid $P^I$ and the lower bid $P^L$ (NO–NO), in this case, WTP $< P^L$

A common issue that researchers face while applying the CV method is the identification and treatment of protest WTP responses [29]. In CV studies, protest responses can account for 50% of WTP [30,31].

The most common treatment of protest bids is the exclusion of them from the sample [31,32]. However, some researchers argue that only deleting is not an option, it is important to investigate protest responses to define the motivation behind protest bids [29,30]. Thus, several reasons have already been identified in the literature. Namely, possible subjects of protest might be (a) need in more information or (b) a conviction that the government is responsible for payment, while (c) "I cannot afford it" is defined as a true WTP of zero [29,30].

### 2.2. Reasoned Action Approach

We use the reasoned action approach (RAA) to assess the level of influence that psychological factors may have on Kazakh citizens' valuation of the government policy aimed at increasing milk production through co-operatives. How psychological factors may underlie individual's behaviour was stated by Fishbein and Ajzen [33] in their theory of reasoned action (TRA), where beliefs, attitudes, intentions, and behaviour were identified as its main elements. The TRA was extended by adding perceived behavioural control in the theory of planned behaviour (TPB) [34], which was defined as a determinant of behavioural intention and behaviour [35]. RAA is a continuation of the TPB, where behaviour is assumed to consist of four elements—action, target, time, and context [36]. Hence, the generality of behaviour can be controlled by making those elements more or less specific. Following the RAA, individuals construct (a) behavioural belief $b_i$, which is weighted by evaluation $e_i$ of its outcome, (b) normative beliefs $n_i$ that are evaluated by the motivation to comply $m_i$ with a referent, and (c) control beliefs $c_i$ assessed by the power $p_i$ of that belief. Together they compose attitude (i.e., $A = \sum b_i e_i$), social norms (i.e., $SN = \sum n_i m_i$), and perceived behavioural control (i.e., $PBC = \sum c_i p_i$), which underly the

intention to perform the given behaviour (Figure 1). Thus, constructed and weighted *A*, *SN*, and *PBC* are combined to formulate the behavioural intention (BI).

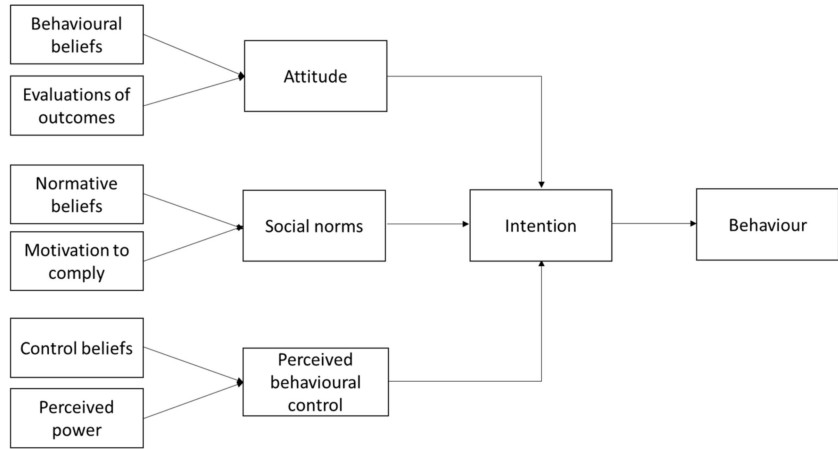

**Figure 1.** The reasoned action approach. The figure was drawn by authors following the model described in the text.

*2.3. Other Constituents of the Model*

We expand the RAA framework to include other contextual elements that may be relevant in the respondent's valuation of the policy in our framework (Figure 2).

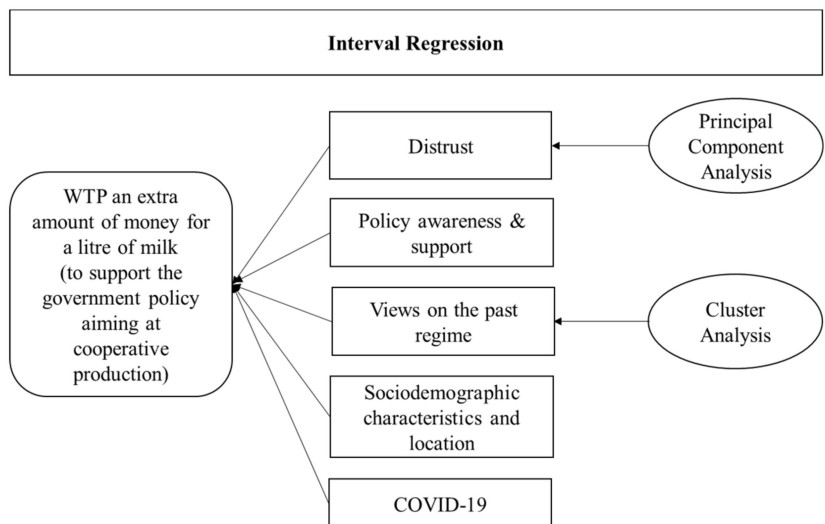

**Figure 2.** The conceptual framework of the study.

Prior to announcing independence in 1991, Kazakhstan was a part of the Soviet Union and regarding the collectivist–communist regime, agricultural production was organised mostly on the basis of collective farming, i.e., kolkhozes and sovkhozes [6,37,38]. Even though almost 30 years have passed since the collapse of collective farms, the transition from centrally planned to market economy may have left some impact on individuals' views towards the current government and its policies. Although numerous studies tried to shed a light on implications of the transition economy on post-Soviet countries' development [39–42], the influence of post-communist regime on the policy in question is not yet clear. Thus, we investigate how individuals' views on the past regime may be associated with their valuation of a policy aimed at increasing production co-operatives. Several main associations are possible. Individuals who miss the Soviet Union may (a) be supportive of the policy that reminds them of the previous regime (the structure and

function of kolkhozes as agricultural production systems), but they may also (b) be sceptical about the current regime delivering the policy on co-operative production as one in the past, and as a result, may be less likely to support it. Thus, the mistrust of the current regime and unattractiveness of current policies compared with the Soviet Union regime might lead to less support of the current regime by the general public.

Moreover, we investigate the association of (a) sociodemographic characteristics, (b) the location where a respondent resides, (c) awareness of the policy in question, and (d) COVID-19 relationship with respondents' intention to pay extra money for a litre of milk.

### 2.4. Survey and Questionnaire

A snowball sampling technique was used to contact Kazakh citizens to voluntarily take part in the study, i.e., by using an already existing network of contacts via social media platforms to distribute the link to the questionnaire. The questionnaire was approved by the Ethics Committee of the University of Reading (protocol code/ethical clearance application number 001151P, approved on 2 December 2019).

The instrument used to collect information was a questionnaire survey using Qualtrics XM (Version 12, Qualtrics, Provo, UT, USA). All participants were provided with an information sheet and consent form containing information about the aims and objectives of the research. The questionnaire was created in English and translated to Kazakh and Russian. To guarantee accuracy, a second, independent person reviewed and edited the translation for accuracy, natural flow in the target language, and adherence to the needs of the survey.

The data were collected in two periods, before and during the COVID-19 pandemic. The first wave of data collection ($n = 272$) was completed in a month period, between 10 December 2019 and 10 January 2020.

In March 2020, the first case of COVID-19 was reported in Kazakhstan and the government implemented a lockdown for two months, until May 2020. However, as soon as the restriction was eased, the number of cases of the disease increased sharply, reaching its peak in June–July 2020. Considering the situation and the government's measures to deal with it, in June 2020, we took the opportunity of exploring the effect of COVID-19 on respondents' WTP. Therefore, during the period of a month, between 13 June and 13 July 2020, 234 fully complete additional responses were collected, making a total of 506 observations.

The questionnaire consisted of five sections (awareness and support, CV, RAA, views on the past regime, sociodemographics, and location) and included a total 37 questions.

The aim and features of the governmental policy were delivered in the form of short informative text within the first section of the survey and respondents were asked to respond (a) if they have had information about co-operative creation and (b) if they agreed with the aim of the policy.

Within the CV section, respondents were asked to answer the WTP questions. During the pilot study in August 2018, we used open-ended questions allowing respondents to decide without giving options, then received an amount of money that was used to adjust main bids for WTP. Information from the pilot questionnaire was used to assign the prices for the WTP questions (KZT 10, 40, 70, 100, and 130). Thus, the amount of money Kazakh citizens are willing to pay for the transformation in the dairy sector was obtained by providing information about the governmental policy and asking them the following question: "Would you be willing to pay extra X amount of money for a litre of milk in order to support the government's policy?" where X amount of money was chosen randomly from the given bids.

If respondent answered "No", then the requested amount of money was decreased by KZT 15 ($P^L$) or it was increased to KZT 15 ($P^H$) if the answer was "Yes".

If a respondent ticked the fourth option and answers No–No, then further questioning was used to indicate the reasons. The third section of the questionnaire included questions on RAA in order to reveal psychological aspects underlying Kazakhs citizens' intention

to pay an extra amount of money for a litre of milk. Salient beliefs of the respondents were defined during the pilot study in November 2019 by asking open-ended questions towards the support of the governmental policy aimed at co-operative creation; following that, the statements were identified and included in the survey. Respondents were asked to rate the RAA statements on a set of unipolar and bipolar evaluative adjective scales, with five places. To elicit attitude (A) toward paying an extra amount of money for a litre of milk in order to support the government policy, for instance, respondents were asked to score the strength of belief about a consequence of the behaviour from 1 to 5 (i.e., extremely unlikely–extremely likely), while evaluation of the belief was assessed from −2 on the negative side to +2 on the positive side. Thus, the higher the behavioural belief the more it was expected to have a positive influence on attitude. Consequently, the sum across all scales (since there are three behavioural outcomes, the possible range of the scale for A is from −30 to +30) was taken as a measure of a respondent's attitude towards co-operative production. The same procedure was applied to reveal SN and PBC with some differences on scoring, namely, (a) respondent's normative beliefs were scored from −2 to 2 (i.e., extremely unlikely–extremely likely), while the motivation to comply with a referent took on values from 1 to 5; (b) control beliefs were scored from 1 to 5, while the power (P) of the factor was scored from −2 to +2 on statements capturing facilitating factors (i.e., P1) and from 2 to −2 on statements capturing impeding factors (i.e., P2, P3, and P4) [36]. Hence, the scale for the SN and for the PBC ranged from −40 to +40.

Table 1 shows statements used to reveal Kazakh citizens' A, SN, and PBC. During the survey, prior to responding on RAA questions, respondents were informed about the aim and features of the governmental policy in the form of short informative text.

**Table 1.** Statements to reveal respondent's attitude, social norms, and perceived behavioural control towards the behaviour.

| Item | Questionnaire Statements | Scale |
|------|--------------------------|-------|
| | **Attitude** | |
| B1 | Paying an extra amount of money for a litre of milk would improve the quality of milk | extremely unlikely–extremely likely |
| E1 | For me improving of the quality of milk is | extremely bad–extremely good |
| B2 | Paying an extra amount of money for a litre of milk would motivate farmers to produce better | extremely unlikely–extremely likely |
| E2 | For me motivating farmers is | extremely bad–extremely good |
| B3 | Paying an extra amount of money for a litre of milk would support domestic milk production | extremely unlikely–extremely likely |
| E3 | For me increasing domestic milk production is | extremely bad–extremely good |
| | **Social norms** | |
| N1 | My spouse/partner thinks that it would be good for me to pay an extra amount of money for a litre of milk | extremely unlikely–extremely likely |
| M1 | With regards paying an extra amount of money for a litre of milk, I want to do what my spouse or partner thinks I should do | strongly disagree–strongly agree |
| N2 | My close relatives think that it would be good for me to pay an extra amount of money for a litre of milk | extremely unlikely–extremely likely |
| M2 | With regards paying an extra amount of money for a litre of milk, I want to do what my close relatives think I should do | strongly disagree–strongly agree |
| N3 | My parents think that it would be good for me to pay an extra amount of money for a litre of milk | extremely unlikely–extremely likely |
| M3 | With regards paying an extra amount of money for a litre of milk, I want to do what my parents think I should do | strongly disagree–strongly agree |
| N4 | My close friend thinks that it would be good for me to pay an extra amount of money for a litre of milk | extremely unlikely–extremely likely |
| M4 | With regards paying an extra amount of money for a litre of milk, I want to do what my close friend thinks I should do | strongly disagree–strongly agree |

**Table 1.** *Cont.*

| Item | Questionnaire Statements | Scale |
|------|--------------------------|-------|
| | **Perceived behavioural control** | |
| C1 | I have enough money to pay an extra amount of money for a litre of milk | extremely unlikely–extremely likely |
| P1 | Having enough money would make it easier for me to pay an extra amount of money for a litre of milk | strongly disagree–strongly agree |
| C2 | I don't trust dairy factories to pay an extra amount of money for a litre of milk | extremely unlikely–extremely likely |
| P2 | The lack of trust in dairy factories would make it difficult for me to pay an extra amount of money for a litre of milk | strongly disagree–strongly agree |
| C3 | I don't trust farmers (households) to pay an extra amount of money for a litre of milk | extremely unlikely–extremely likely |
| P3 | The lack of trust in farmers (households) would make it difficult for me to pay an extra amount of money for a litre of milk | strongly disagree–strongly agree |
| C4 | I don't trust the government's policy to pay an extra amount of money for a litre of milk | extremely unlikely–extremely likely |
| P4 | The lack of trust in the government's policy would make it difficult for me to pay an extra amount of money for a litre of milk | strongly disagree–strongly agree |

The statements "During the Soviet Union people had more healthy food"; "During the Soviet Union Kazakhstan's economy was better"; and "I like the idea of collective farming (kolkhozes) during the Soviet Union" in section 4 of the questionnaire were used to capture whether the respondent's views on the past regime are associated with their willingness to support the governmental policy.

Finally, age, education, gender, and income composed the sociodemographic part of the survey. Within this part, respondents were also asked to indicate the location where they reside.

### 2.5. Statistical Analysis

The analysis comprised a combination of quantitative methods including cluster analysis on the respondent's views on the Soviet Union (SU) and parameter model estimation using an interval regression model.

### 2.5.1. Cluster Analysis

Cluster analysis is used to group respondents according to their views on the past regime. Concisely, it involves a search through data for observations that have high similarity in comparison to one another but are very dissimilar with respect to objects in other clusters.

Two main approaches are known to cluster analysis: hierarchical and partitioning. Considering the hierarchical approach, which can also be interpreted as a top–down procedure, each observation represents its own cluster. At any following stage, similar and closer in characteristics clusters merge, creating a group and continue until cutting the tree at a suitable level. Otherwise, the procedure terminates when all members of a group are consistent, creating one common cluster at the top of a tree-like form, called a dendrogram [43–45].

In the partitioning (*k*-means) approach, a cluster can be formed by specifying the number of clusters prior to the analysis. Using this number as an input, the algorithm specifies an initial centre of the cluster (i.e., *k*), afterwards, observations are assigned to the cluster according to their nearest cluster centres (i.e., one of the *k* clusters). According to the *k*-means approach, the number of clusters is not known in advance [43–45]. Therefore, the choice of an initial configuration can be based on the results of hierarchical clustering [46]. Since *k*-means is stated as superior to the hierarchical methods due to its ease of implementation, simplicity, efficiency, and empirical success [44,46], we followed this approach. Thus, initially, the number of clusters was identified through the dendrogram, and then the *k*-means method was applied.

### 2.5.2. Interval Regression

An interval regression model, a generalisation of the Tobit model [47], was used to analyse factors underlying Kazakh citizens' WTP extra amount of money for a litre of milk in order to support the government policy aimed at dairy production and creating co-operatives. The singularity of this model is in the observed range of the dependent variable being censored, since the dependent variable $y_i^*$ (i.e., respondent's WTP an extra amount of money for a litre of milk) is unobserved [48]. What is observed is an interval, which has lower $m_i$ and upper $M_i$ bounds,

$$m_i \leq y_i^* \leq M_i \tag{2}$$

where, basically, the data can be defined with three possible outcomes. In the case if the lower bound is known, but the upper is not, then "right-censored"; or vice versa, if the upper bound is known, but the lower is not, then "left-censored". If both lower and upper bound are known, then the data can be defined as an "interval" [49]. We can state that

$$y_i^* = x_i \beta + u_i, \ u_i | x_i \sim Normal\left(0, \sigma^2\right) \tag{3}$$

where $x_i$ is a vector of an explanatory variable of WTP of a respondent $i$ and $\beta$ is a parameter vector associated with explanatory variables $x_i$. These are the RAA variables (attitude, social norms, and perceived behavioural control), cluster variable accounting for respondents who like the past regime, policy awareness, sociodemographic variables (age, education and income), and location. The error term $u_i$ is assumed to be normally distributed with mean zero and standard deviation $\sigma$ [50,51].

## 3. Results and Discussions

### 3.1. Descriptive Statistics

Descriptive statistics of the explanatory variables are shown in Table 2. Lower and upper are dependent variables, which refer to left-censored and right-censored observations. A, SN, and PBC were generated following [36] (see Section 2.2). Two variables were created to indicate the awareness (i.e., infopolicy) and support (i.e., policyagree) of the considered policy, respectively. SU_likers is an explanatory variable obtained from the cluster analysis and captures respondent's views on the past regime, taking a value of 1 for those with a relatively positive view on the past regime and 0 otherwise. A dummy variable for COVID-19 was created with a value of 1 for respondents participating during the COVID-19 wave and 0 otherwise.

Finally, sociodemographic variables including age, education, gender, income, and location are the explanatory variables that refer to the sociodemographic and location part of the study. Almost 60% of the respondents were female. Nearly 50% belonged in the age band of 18–30, and up to 80% were aged below 50 years old. A quarter had education at school and college level, while undergraduate and postgraduate levels of education were 43% and 30%, respectively. Almost 40% of the respondents stated their income up to KZT 100,000, which can be defined as low income, about 25% indicated middle income (KZT 101,000–150,000), while the remaining 35% were respondents with high income. The majority of respondents reside in the capital (about 68%), while the rest were from different cities. Therefore, within the location variable, we treated the capital as a zero point and identified the distance to other cities in kilometres from the capital.

A comparison between the Kazakh population in 2019 and our survey sample is provided in Table 3.

The main difference is education at school and college level, and household income up to KZT 50,000 being underrepresented, while education at postgraduate degree and household income over KZT 100,000 are overrepresented. Education and level of income are highly correlated to one another, and since the survey was distributed mainly with the support of colleagues from national universities, the sample covered mostly educated and

high-income earning respondents. Although most of the population hold the average per capita income of up to KZT 100,000, the sample household income was equally distributed amongst the 4 categories.

**Table 2.** Variable definitions and statistical descriptions.

| Variable | Definition | Mean | Min | Max |
|---|---|---|---|---|
| Lower | Obs. (*n* = 284), lower bound | 67.757 | 0 | 145 |
| Upper | Obs. (*n* = 157), upper bound | 66.382 | 0 | 145 |
| A | Attitude of the respondents towards the co-operative creation policy | 15.991 | −13 | 30 |
| SN | Perceived social norms of the respondents | 7.126 | −34 | 40 |
| PBC | Perceived behavioural control of the respondents | −4.009 | −40 | 24 |
| SU_likers | cluster derived by the cluster analysis; dummy variable 1 = like the Soviet Union regime; 0 = otherwise | 0.586 | 0 | 1 |
| infopolicy | dummy variable, 0 = if otherwise; 1 = if the respondents received information about the government policy before; | 0.233 | 0 | 1 |
| policyagree | dummy variable, 0 = if otherwise; 1 = if the respondents agree with the aim of the policy | 0.926 | 0 | 1 |
| age | Age of the respondents 1 = 18–30; 2 = 31–49; 3 = 50 and older | 1.733 | 1 | 3 |
| education | The final completed education of the respondents 1 = school; 2 = college; 3 = undergraduate; 4 = postgraduate | 2.932 | 1 | 4 |
| gender | dummy variable, 0 = male, 1 = female | 0.623 | 0 | 1 |
| income | The respondent's monthly income 1 = KZT 0–50,000; 2 = KZT 51,000–100,000; 3 = KZT 101,000–150,000; 4 = KZT 151,000 and higher | 2.797 | 1 | 4 |
| location | the location of the respondents in kilometres from the capital | 296.877 | 0 | 2600 |
| covid | dummy variable, 0 = pre-COVID-19 period, 1 = COVID-19 period | 0.592 | 0 | 1 |

**Table 3.** Socioeconomic characteristics of Kazakhstan population (2019), percentage of Kazakhstan population versus percentage of the sample.

| | Number of Individuals | Kazakhstan Population (%) | Sample, *n* = 326 (%) |
|---|---|---|---|
| Total population | 18,395,567 | — | — |
| Female population | 9,749,650 | 53 | 62 |
| Male population | 8,645,916 | 47 | 38 |
| Age (15–34, Kazakhstan; 18–30, sample) | 5,509,210 | 42 | 46 |
| Age (35–54, Kazakhstan, 31–49, sample) | 4,504,423 | 35 | 35 |
| Age (55+) | 3,034,521 | 23 | 19 |
| School | 117,204 | 28 | 10 |
| College | 144,333 | 34 | 17 |
| Undergraduate | 142,435 | 34 | 43 |
| Postgraduate | 22,765 | 5 | 30 |
| Household income (<KZT 50,000) | n/a | 50 * | 15 |
| Household income (KZT 51,000–100,000) | n/a | 39 * | 25 |
| Household income (KZT 101,000–150,000) | n/a | 8 * | 25 |
| Household income (>KZT 151,000) | n/a | 3 * | 35 |

Note: Figures for the level of education of the population are based on the number of individuals who finished each of the education categories during 2019; * distribution of population by average per capita income (by the number of the population is not available). An average nominal per capita income of the population was KZT 104,282 in 2019. The data were derived from the official website (www.stat.gov.kz, accessed on 10 January 2021) of the Statistics Committee of the Republic of Kazakhstan.

### 3.2. Cluster Analysis

Overall, three statements were used to define the views of respondents towards the past regime. Respondents were asked to evaluate these statements from strongly disagree to strongly agree on a 5-point Likert scale. Primarily, we conducted a hierarchical procedure for these variables to determine the number of clusters by using the dendrogram. Then, we checked the validation of the chosen number through Calinski and Harabasz's

and Duda–Hart indices (i.e., cluster stopping rules). Both indices showed $n = 2$ cluster as appropriate.

Once the number of clusters was specified, a *k*-means procedure was carried out. Table 4 illustrates the summary statistics of the clusters by means. Cluster 2 was characterised by having higher mean rates, while cluster 1 had mean = 3 or less on the given statements. Therefore, cluster 2 is assumed that it captured the Soviet Union regime likers, while cluster 1 is not. We created a dummy variable with a value of 1 for SU_likers and a value of 0 otherwise (non-SU_likers).

**Table 4.** Summary statistics (by mean) of the clusters.

| | During the Soviet Union People Had More Healthy Food | During the Soviet Union, Kazakhstan's Economy Was Better | I Like the Idea of Collective Farming (Kolkhozes) during the Soviet Union |
|---|---|---|---|
| 0 = non-SU_likers (Cluster 1) | 2.978 | 2.000 | 2.467 |
| 1 = SU_likers (Cluster 2) | 4.654 | 3.702 | 3.974 |
| Total | 3.960 | 2.997 | 3.350 |

### 3.3. The Value of the Policy for Society

The average premium price of the respondents WTP for a litre of milk to support the policy was KZT 103. The average market price paid by respondents for a litre of milk in the period of the study was KZT 300. This means that on average respondents are prepared to pay 34% more than the market price to support the policy in production co-operative creation. However, this is possibly an overestimate given that our sample contains more respondents with relatively high levels of income. For the purpose of obtaining a WTP estimate that is more representative of the population, we looked at how the WTP varies according to sociodemographic characteristics (Table 5). Using the household income population information (Table 3), we weighted the estimated WTP by income group according to the population (%) in each income group. This gives the WTP of KZT 86.61 (i.e., a 29% premium price).

**Table 5.** The estimated average WTP according to sociodemographic characteristics of the respondents.

| | Obs. | Mean | S.D. |
|---|---|---|---|
| Female population | 203 | 100.32 | 40.60 |
| Male population | 123 | 106.88 | 41.24 |
| Age (18–30, sample) | 149 | 105.34 | 40.84 |
| Age (31–49, sample) | 115 | 100.34 | 39.62 |
| Age (50+, sample) | 62 | 101.24 | 43.64 |
| School | 32 | 109.15 | 47.05 |
| College | 56 | 100.46 | 44.83 |
| Undergraduate | 140 | 107.90 | 41.29 |
| Postgraduate | 98 | 94.75 | 34.59 |
| Household income (<KZT 50,000) | 50 | 77.49 | 32.17 |
| Household income (KZT 51,000–100,000) | 81 | 89.27 | 41.08 |
| Household income (KZT 101,000–150,000) | 80 | 121.89 | 40.02 |
| Household income (>KZT 151,000) | 115 | 110.04 | 36.21 |

The budget of the program, where the creation of co-operatives had been stated, was 2374.2 billion tenges (KZT) for five years (i.e., 2017–2021). We highlight that the program covered not only the support of small farmers through creating co-operatives but also other sectors, including (a) efficient use of water and land resources; (b) increasing the provision of agricultural producers with equipment and chemicals, and (c) scientific–technological, personnel and information–marketing support of the agroindustrial complex.

Once the individual average WTP for the policy is estimated, we can use it to estimate the economic value of the policy in a relatively simple way. Assuming that to evaluate

the policy, a certain age needs to be reached, the total value of the policy was calculated by multiplying the number of Kazakh citizens at age 15 and over (13,000,000) (Table 3) by the corrected average WTP (i.e., KZT 86.61) times % Kazakh population consuming milk (approximately 90% of the population): kg milk/dairy consumed per month (22 kg) times 12 months. Then the estimate for the total economic value of the policy aiming at the creation of the co-operatives for the Kazakh citizens is KZT 267 billion per year, or KZT 1335 billion per five years (the five-year Program period), which is half of the total budget for the whole program. The economic value of the policy would equal the cost of the whole program after 10 years.

### 3.4. Drivers for WTP

Table 6 shows how elements of the RAA are associated with respondents' WTP. Namely, attitudes, social norms, and perceived behavioural control are associated with an increase in participants' WTP an extra amount of money for a litre of milk in order to support the government policy ($p$-values < 0.01). These results are in line with studies on consumer's willingness to purchase organic milk [52], to purchase pasture-raised livestock products [53], and to pay for meat from mobile slaughter units [54]. In other words, if the attitude towards the behaviour (i.e., paying a premium price for a litre of milk to support the policy) is more positive than negative, it is more likely that the behaviour will be performed. Furthermore, if other people (i.e., spouse/partner, close relatives, close friends, and parents) who are considered highly important by the individual are believed to approve rather than disapprove and also perform this behaviour, people are more likely to feel social pressure to engage in this behaviour. Additionally, following the model and the results of the study, if Kazakh citizens perceive more facilitating than inhibiting factors, perceived behavioural control should be high, consequently the behaviour will be performed.

**Table 6.** Results of the interval regression.

| | Coefficient | z-Statistics |
|---|---|---|
| A | 1.34 *** | 2.59 |
| SN | 1.13 *** | 3.19 |
| PBC | 1.22 *** | 2.95 |
| 1. SU_likers | −33.90 *** | −3.60 |
| 1. Infopolicy | 24.72 ** | 2.37 |
| 1. policyagree | 9.88 | 0.63 |
| **Age (18–30, base category)** | | |
| 31_49 | −15.23 | −1.51 |
| 50 and older | −28.81 ** | −2.17 |
| **Education (School, base category)** | | |
| College | −9.50 | −0.52 |
| Undergraduate | −12.95 | −0.73 |
| Postgraduate | −32.59 * | −1.71 |
| 1. female | 1.86 | 0.19 |
| **Income (<KZT 50,000, base category)** | | |
| KZT 51,000–100,000 | 6.78 | 0.49 |
| KZT 101,000–150,000 | 49.76 *** | 3.35 |
| >KZT 151,000 | 34.62 ** | 2.40 |
| Location | 0.02 ** | 2.17 |
| 1. COVID-19 | −26.20 *** | −2.79 |
| _cons | 94.30 *** | 3.85 |
| sigma | 62.37 | 14.78 |
| Number of observations | 326 | |
| Left-censored | 42 | |
| Right-censored | 169 | |
| Interval-censored | 96 | |
| Log-likelihood | −488.23 | |
| LR chi2(17) = 83.93; Prob > chi2 = 0.0000 | | |

Note: *, **, *** for 10, 5, and 1% of significance level, respectively.

The results also show that Kazakh citizens who like the Soviet Union regime were willing to pay KZT 33.90 (1 US dollar (USD) is equal to 426.84 tenges (KZT) as of 27 June 2021) less to support the policy on production co-operatives creation than citizens who do not like the Soviet Union regime ($p$-value < 0.01). Possible reasons for this result may relate to the possibility that individuals who like the Soviet Union (i.e., who perceive the past Communist as a better regime than the current regime) may also have a feeling of frustration with democracy [55]. Moreover, one of the reasons behind satisfaction with the past regime was its stability and guarantee of basic needs [55]. As pointed out by Toleubayev et al. [56], "Kazakhstani people express great nostalgia for their past lives in the Soviet era and their narratives express a strong appreciation for the level of social security, income stability, low food prices, and the sense of a more egalitarian communal life". This frustration present in post-communist countries may be consequence of a transition economy towards a "wild capitalism" characterized by "rapid and massive liberalization, by the lack or the inefficiency of the state intervention in the economy, by corruption, and significant social movements of protest", and not achieving the similar level of democracy such as in Western Europe [57,58].

The lower support for the policy on production co-operatives creation by Kazakh citizens who like the Soviet Union is reinforced by the finding that people aged over 50 are less supportive of the policy (Table 6). Hence, results suggest that Kazakh citizens with a positive attitude towards the old Soviet Union regime, and aged over 50, are more likely to perceive policies from the new regime (since independence) as unattractive and ineffective.

The results indicate that respondents' WTP is positively associated with having adequate information about the policy ($p$-value < 0.05). Kazakh citizens with relatively higher awareness about the policy are ready to pay about KZT 25 (1 US dollar (USD) is equal to 426.84 tenges (KZT) as of 27 June 2021) more than those who had no knowledge before. Undoubtedly, for a respondent receiving essential information about the product may be crucial for decision making. A similar finding was also reported by Stampa et al. [53] and Zhang et al. [14]. Moreover, Zhang et al. [59] found that increasing awareness of cultured meat influenced positively on Chinese consumer's acceptance of it. A similar effect was found by Roosen et al. [60], when investigating consumers' WTP for nanotechnology food differed according to the information provided.

The results showed an increase in income is associated with a higher WTP. Respondents with the income between KZT 101,000 and KZT 150,000, and more than KZT 151,000 are willing to pay KZT 50 and KZT 35 (1 US dollar (USD) is equal to 426.84 tenges (KZT) as of 27 June 2021) more, respectively, than respondents with monthly income up to KZT 50,000. This finding is expected and in line with [13,61], where a WTP was stated being increased with higher levels of income.

Although the respondents holding postgraduate level of education are less likely to support the policy ($p$-value < 0.10), the reason for this is unclear. However, it is noted that the share of highly educated respondents was higher in the sample of the study.

The location is found to be statistically significant ($p$-value < 0.05), and thus, individuals living apart from the capital are more inclined to pay a premium price for a litre of milk to support the policy. This is justified since the policy is oriented for the development of the rural areas and Kazakh citizens' living in regions (apart from the capital) perceive more the importance of the policy.

The parameter measuring the relationship between COVID-19 and respondents' WTP was found to be statistically significant ($p$-value < 0.01) suggesting that COVID-19 might have had some impact on individual's WTP. Kazakh citizens seem less likely to support the government policy on creating co-operatives under the COVID-19 situation. Results show that individuals average WTP for the government policy aimed at increasing the number of co-operatives was lower during the pandemic period compared to the pre-pandemic period. Thus, the average WTP to support the policy was KZT 118 (1 US dollar (USD) is equal to 426.84 tenges (KZT) as of 27 June 2021) prior to COVID-19 outbreak, whereas during the pandemic it decreased by 22% and was KZT 92. This can be due to the rise of

unemployment [62], stated as one of the dramatic implications of the COVID-19, which touched Kazakhstan as well. According to the news agency "Khabar 24" [63], during the pandemic, the number of unemployed Kazakh citizens only in one city has increased by 3.5 times. Thousands of entrepreneurs were forced to pause their work; about 1.6 million employees were sent to leave without payment. Thus, widespread dissatisfaction with the measures taken by the government to stop the spread of the virus might cause decreased support of the current government by the general public.

### 3.5. Protest WTP Responds

Within *n* = 506 observations, *n* = 180 were labelled as protest bids and deleted, which is almost 35% of the sample.

Respondents were asked to state the reason for zero WTP, where the most common four reasons are found. Both "I am already paying tax and think that the government has to use that money to support" and "The prices of milk/dairy products are already expensive" were stated 67 times. Next was, "I am sceptical about that the money will go to the farmers" that was repeated in 52 places; 45 times protestors mentioned, "I will need to have more information about this policy". Although "I don't have enough income to pay extra money" was stated 56 times, this reason was labelled as true WTP of zero, therefore were not excluded from the sample.

### 3.6. Policy Implications

Our results show the readiness of the general public to support the government's plan in creating production co-operatives and the economic viability of the plan. However, it is important to acknowledge that the success of the policy also depends on the rural households' willingness to participate in the policy. Kaliyeva et al. [16] revealed the existing interest of rural households in joining and creating co-operatives in Kazakhstan. Hence, policies aimed at the creation of co-operatives can be a viable solution to increasing milk production in Kazakhstan. It is worth noting that the government could also take other approaches to increase dairy/milk production. For instance, policies such as promoting family farming by introducing tax relief and/or subsidies could also achieve the aim of increasing milk production, but farmers would not have the same level of access to information and technology that a co-operative would offer. The level of public support for policies promoting family farming is unknown, but this policy may find less opposition from individuals liking the SU.

The policy on co-operative creation might facilitate connection of farmers (rural households) with supply chains (dairy factories). Not only producers (farmers, dairy factories) might benefit from the policy, but also society. It is acknowledged that co-operatives can help developing local value chains as well as facilitate the access to local and global markets [64]. The structural changes in the dairy sector may enhance the production of domestic products, and as a result may positively affect the country's trade balance by reducing the demand on imported dairy products. Moreover, co-operatives are an acknowledged way of reducing poverty in rural areas and enhancing sustainable development [8–10].

Considering research findings in other countries, there are two points worth discussing: (a) what kind of co-operatives can help competitiveness in agriculture and (b) what has been the experience of policies supporting the creation of co-operatives. It is worth noting that research conducted in other countries on agricultural co-operatives is diverse and provides useful information to understand how regional characteristics/conditions may influence the potential effects of creating co-operatives on agricultural production and markets. The creation of co-operatives among enterprises in direct competition with each other allow producers to take advantage of synergies and reinforce bargaining power without major losses of freedom or flexibility [65]. This may be particularly important in developing countries where the size of the farming system is small. Li and Ito [66] show that agricultural co-operatives in regions where agricultural land size is relatively small (e.g., China) can help in developing other markets associated with agricultural production

(e.g., development of land rental markets by reducing transaction costs). Liang [67] argues that producer co-operatives act as a competitive yardstick of markets leading to competitive markets. Liang [67] also shows that this yardstick effect resulted into higher farm gate prices for hog producers in China. In addition, the yardstick effect may lead to a reduction in production costs [67].

Co-operative and community-based forms of doing agriculture are common in most countries, especially in developed countries where "the access of small farmers to markets is usually facilitated by agricultural service co-operatives" [68]. According to recent research, 134 agricultural co-operatives in the US celebrated their 100th anniversary in 2014 [69]. Research on the longevity of agricultural co-operatives in developed countries listed several main reasons for that, such as the achievement of scale economic gains and the ability to adapt to dynamic situations. The success or failure of policies supporting the creation of co-operatives may depend on the existing institutional conditions as well as in the level of trust on the government by producers and the degree the regulatory policy with too regulatory policies being less likely to succeed, particularly in post-Soviet countries [70]. Research on the success and failure cases of agricultural co-operatives in developing countries revealed the lack of comprehensive support, including advice on best practices and monitoring co-operative activities as the main reasons for the failure of banana co-operatives in Rwanda [71]. Moreover, Moon [71] suggested that the success of the creation of co-operatives might be possible through the efforts of both the aid agency and the beneficiaries.

Although what share of the total budget was aimed to be used for co-operatives creation is not clear, the results of the study showed the importance of the policy for the Kazakh society. Extrapolating to the Kazakh population who consume milk/dairy products would mean that the economic value of the policy would be KZT 1335 bn for the length of the program at KZT 267 bn per year, which is approximately half the total program budget, and includes other interventions beyond the creation of co-operatives. The economic value of the policy would equal the cost of the program after 10 years. This indicates there is public support for this policy.

Our findings suggest that although there is general support for the policy, there are still parts of the population, i.e., individuals missing the SU regime, who may mistrust newly created organisational forms of the current government. Therefore, as a country with a transition economy, the Kazakhstan government may face nonacceptance of the policy by some of the population. The main reason is found to be the implications of the wild capitalism that Kazakh people faced after the transition from communism to a market economy. Public rejection of the policy might also be connected with COVID-19, which had dramatic damage to the economy of the country. Therefore, the government attempts for increasing its attractiveness will lead the policy to be more widely supported.

Provision of information about the policy (e.g., aims, implementation) was found to be important in respondents supporting the policy. We therefore recommend that policymakers need to resolve any unambiguity in definitions of the use of the term "co-operative" under the current policy, "that will prevent any possibility of misunderstanding or misinterpreting the strategic intentions" [68]. Hence, in order to gain policy support for increasing dairy/milk production by creating co-operatives, good communication of the policy seems key to building trust amongst Kazakh citizens. Finally, a "top–down" route to the creation of agricultural co-operatives has been widely criticized around the world due to its nonviability and noneffectiveness [68]. Survey results in this research showed that information on the policy aimed at creating co-operatives had neither been widely distributed nor explained to Kazakh citizens and rural households [16]. The majority of the participants only discovered the existence of the program from the researchers during the survey. However, in the developing world the "top–down" process can be a legitimate way of organising co-operatives [72]. For instance, the classic form of establishing co-operatives in China that involves the participation of the state and farmers has been regarded as widespread and effective. In post-socialist Vietnam, state involvement also played a crucial

role in the development of agricultural co-operatives, where the sector suffered from low levels of initiative on the part of farmers [72,73]. Despite this, we believe that the initiative to create co-operatives should come from rural households. Moreover, dairy factories need to also be involved in such initiatives from the outset. Otherwise, the top–down process may not be implemented successfully.

## 4. Conclusions

We assessed the public support for a policy aimed at increasing milk production through co-operatives by estimating the monetary value for society of the policy. It was found that Kazakh citizens showed support for the government policy. The findings presented in this paper might also be relevant for post-communist countries, such as Russia, Ukraine, and Kyrgyzstan, the agricultural development of which has a similar pattern to Kazakhstan's.

Psychological factors played an important role in the success of the policy—namely, holding a positive attitude towards the behaviour, having positive endorsement regarding the behaviour (the support of the policy) from the social referent (e.g., family members and friends), and being in a position to control the behaviour, i.e., A, SN, and PBC, significantly influence Kazakh citizens' WTP support of the policy. Moreover, individual awareness of the policy was found to be important in supporting the policy. Therefore, good communication of the policy and its aims to the general public is key for policy support. Findings suggest that countries that have transitioned to new policy regimes can face difficulties in implementing policy programmes in cases where significant parts of the population miss characteristics of the past regime. We also found some evidence of reprioritisation of people's preferences under COVID-19, with relatively lower support for the policy. Therefore, to achieve the support of the general public, the government should take measures to increase its attractiveness and try to earn public acceptance.

In this study, we investigate the success of the Kazakh government policy aimed at increasing milk production through an increase in co-operative production. We mainly based our analysis on the opinion and reactions of the general public in Kazakhstan. However, other policy outcomes, such as an increase in the competitiveness of Kazakh's milk production in international markets, could also generate further benefits (e.g., extra government revenue). In addition, accounting for any environmental effects (e.g., landscape and habitat, biodiversity, soil) associated with a change from current production to co-operative production would also be needed in a cost–benefit analysis.

Additionally, it should be emphasized that this research considered only a single attribute, i.e., the value of the policy on creation of production co-operatives. However, there is a potential for exploring the general public's willingness to pay for co-operatives through including other specifications. These might include other attributes, including diversity of co-operatives such as service co-operatives. Alternatively, consumers' preferences can be explained by extending product attributes, e.g., quality and price of the milk from co-operatives. In such a case, a choice experiment approach can be utilized to investigate individuals' WTP for welfare changes by offering different attributes of goods/policies and choosing a preferred option across several sets [74,75].

**Author Contributions:** Conceptualization, S.K., F.J.A. and Y.G.; methodology, S.K., F.J.A. and Y.G.; software, S.K., F.J.A. and Y.G.; formal analysis, S.K., F.J.A. and Y.G.; investigation, S.K., F.J.A. and Y.G.; data curation, S.K.; writing—original draft preparation, S.K.; writing—review and editing, F.J.A. and Y.G.; visualization, S.K., F.J.A. and Y.G.; supervision, F.J.A. and Y.G. All authors have read and agreed to the published version of the manuscript.

**Funding:** This study was funded by the University of Reading and the Centre for International Programs (CIP) "Bolashak" of the Republic of Kazakhstan.

**Institutional Review Board Statement:** This study was approved by the Ethics Committee of the University of Reading (protocol code/ethical clearance application number 001151P, approved on 2 December 2019.

**Informed Consent Statement:** Informed consent was obtained from all subjects involved in the study.

**Conflicts of Interest:** The authors declare no conflict of interest.

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
