# Peer review of "Would Kazakh Citizens Support a Milk Co-Operative System?"

_agriculture, doi:10.3390/agriculture11070642_

Round 1
Reviewer 1 Report
KZ dairy coops
Line Comment
192 insert have before left
198 There is not correct; maybe replace by Several
216 were not was
221 reaching not beating
284&289 insert the before k-means
311 remove M. Jeffry
317 Reference error reported
329 insert , before respectively
341 sentence “… 2019 only.” Does not make sense
374 that not to
382 in table the large numbers of digits reported are irrelevant, especially for the SD
393 22 kg (SI requires space)
396 Assertive style but it really sounds like very simplistic economics!
436 uncritically numerous (insignificant) digits are reported in this table
440 who not that
472 insert were before forced
523 for not on
528-9 sentence does not make sense to this reader
550 which not that
553 endorsement not endorse
569 diversity not diversify
Reviewer 2 Report
Agriculture is largely policy-driven in most countries, so the topic, which aims to rate the effectiveness of government policy aimed at increasing the number and share of cooperatives in milk production in Kazakhstan, is very actual and fits the goals of the journal.
It can be agreed that the success of a policy action much depends on the opinion and reactions of stakeholders. The starting point is that the costs of encouraging the creation of cooperatives should be recouped from the extra income generated by consumers' willingness to pay extra for a safer supply of milk and milk products. The consumers WTP determined by several factors, knowing them key to success.
The idea and methodology can be good, but in a cost benefit analysis much more factors should be taken into consideration. One example, how the increasing competitiveness on international market can generate revenue for government which also can cover the cost of this government intervention. The environmental effects also can be taken into consideration. These should be mentioned at least as the limitation of this research.
Readers and the respondent need to know what cooperatives they want to create and how. How are these cooperatives different from those operating in the Soviet system. So the government program needs to be explained in more detail. Not just in terms of content, but when it was introduced and what the results are so far.
We should know is there any other supply chain pathway to reach dairy companies, to increase of efficiency and find common interest. The results also depend on the farmers willingness to join the cooperatives and households motivation of participate in this governmental program.
Despite these shortcomings mentioned here, the chosen methodology serves the purpose well. I would like to point out the use of the Reasonable Action Approach (RAA) to assess the level of influence on psychological factors, which is particularly important for the acceptability or rejection of economic policy interventions in a region with a similar past as Kazakhstan. The conceptual framework of research useful and can be used in other researches on area in question as well.
The questioner and statistical analyses well described. The discussion mainly the policy implication not always on the line of results, these are wider. I miss the international comparison, some fresh literature can be used examining how, and what kind of cooperatives can help the competitiveness of agriculture and what is the success or failure of policies supporting creation of them. Mentioning the examples from other countries of ex-Soviet regions and giving examples from other developing countries and from some developed countries in which the cooperatives mainly marketing cooperatives have long history, would increase the added value of this paper as well. In my mind is that this short comparison can increase the interest of readers, and would be useful for policy makers in Kazashtan as well.
Some other suggestions:
Generally it is a very interesting paper, with high potential. There are places for improvement, The reference list suggested to be extended, The limitation of research should be highlighted.
On page 7 the scale should be explained in the text.
Table 3. Kazakhstan population (2019) versus the sample.
In this Table the data of % column needs to be corrected, the data are not 0, 53, 0.47, but 53, 47. The title and data in the last column of the table should also be corrected as before.
According to the following statement
„Kazakh citizens seem less likely to support government policy under the current circumstances. Results show that an average WTP in the pandemic period was lower compared with the pre-pandemic period” . That is not clear it is valid for supporting for creation of cooperatives in dairy sector or that is a general statement.
The authors use mainly local currency, it would be nice the give the data in US dollar as well, as that is shown in case of export and import.
Generally it is a very interesting paper, with high potential. However there are places for improvement. The reference list suggested to be extended. The limitation of research should also be more highlighted.
Round 2
Reviewer 2 Report
Thank for taking my suggestions into consideration.
I suggest the paper to be published in present form and wish success in the future as well.